# Exploring Adoption and Satisfaction with Self-Service Health Technology in Older Age: Perspectives of Healthcare Professionals and Older People

**DOI:** 10.3390/healthcare10040738

**Published:** 2022-04-15

**Authors:** Lesley Pek Wee Land, Lynn Chenoweth, Yukun Grant Zhang

**Affiliations:** 1UNSW Business, UNSW Sydney, Kensington, NSW 2052, Australia; l.land@unsw.edu.au (L.P.W.L.); yukun_grant.zhang@unsw.edu.au (Y.G.Z.); 2Centre for Healthy Brain Ageing (CHeBA), UNSW Sydney, Kensington, NSW 2052, Australia

**Keywords:** self-service technology, health maintenance, older people, health professionals, customer dominant logic, values

## Abstract

(1) Background. A range of self-service technologies (SST) have been adapted to support the health of older people. Factors involved in older people’s and health professionals’ perceptions of SST in older age were investigated. (2) Methods. Customer Dominant Logic guided this prospective mixed-methods study, including surveys with people 70 years and over and health professionals and individual semi-structured interviews in a sample of survey respondents. Survey data were descriptively analysed, while interview themes were derived inductively. (3) Results. Surveyed (n = 12) people 70 years and over placed higher value, expressed more positive user experience, were more satisfied and had greater recognition of the benefits of SST, compared with (n = 10) health professionals (*p* = 0.001), who considered them to be inferior to traditional healthcare. All seven interviewees agreed that despite accessibility issues and complexity, they valued SST support of older people’s health, thereby confirming the relevance of Customer Dominant Logic in SST offerings. (4) Conclusions. Since older participants were positive and satisfied in using SSTs that are health-supporting, health professionals have a role in encouraging and assisting older people in their use. This requires targeted SST education for health professionals, and more accessible, user-friendly SST and technological support for older people.

## 1. Introduction

Older people contribute much to society, including social and economic contributions through taxation, spending and saving [1]. However, increasing longevity in the world population [2] also poses increased societal challenge and economic burden associated with the costs of healthcare, welfare and supported accommodation for older people [3]. The prevalence of chronic illnesses in older people, who are already heavy users of the health service, places increasing pressure on exploring interventions which may help to alleviate the costs of this problem [4]. Given the high cost of traditional medical management of chronic illnesses, self-service health technologies (SSTs) can present a more efficient and cost-effective alternative [5].

SSTs are frequently employed to complement traditional healthcare services by allowing individuals to self-manage their health and maintain well-being with the oversight of their healthcare practitioner/s [6]. Examples of SSTs include continuous real-time temperature, blood pressure and blood glucose monitoring via data collection points, for example, smart wristbands or door monitors [7]; integrated technologies to enhance communication with family/friends [8] and healthcare professionals [9]; telemedicine for remote regions [10]; optical design systems controlling moving images, colour and light frequencies to create personalised spatial cues [11,12]; hearing aids connected to smart phones [13]; wearable technologies that provide individuals with greater control over their environment [14] and robotic companions [15]. Recent studies show older adults are willing to embrace these forms of user-friendly SSTs as part of overall healthcare support [16], but uptake has been slower than expected [6,17].

Interview and focus group studies suggest there are a number of factors to consider in promoting greater uptake, including the value that health-related SSTs hold for the older person and their health practitioner/s, the person’s confidence, self-efficacy and digital literacy with the technology [18], cognitive and sensory limitations, health issues and physical function [19,20]. For these reasons, some older adults place greater value and have a preference for direct consultation with a health professional than they do with use of SSTs in managing chronic illness [5,6,8,21]. Perceived service quality, in terms of satisfaction with technological applications, has been shown to influence the adoption of online information systems suitable for health support [22]. 

Motivation to embrace technology that includes SST applications is generally influenced by its ‘value’ to the individual, i.e., its functional, emotional and social value [23]. Functional value refers to performance, functionality and utility of the SST, and is the strongest driver of continued use [23]. Emotional value is associated with various affective states aroused by the particular SST, which can be positive (e.g., confidence and pleasure) or negative (e.g., annoyance and fear), and the promotion or alleviation of them. When an individual acquires enough perceived expertise in using and advocating the use of a SST among peers, e.g., smart watch, this conveys social value by achievement of a desired goal such as opinion leader or early adopter of SST [16]. The value that the older person places on the SST will either increase or decrease their satisfaction with the technology which, in turn, may lead to behavioural intentions to engage, or disengage, with the SST in the future [24]. 

As previously identified, the value that a health professional places on SST for specific aspects of health support will also likely influence their promotion of such technology in older people [25]. Nevertheless, where a healthcare professional considers SST is suitable for use by an older person, e.g., electronic blood glucose or heart function monitoring, they also need to take into consideration the older person’s values regarding the particular SST function [26,27]. It is unclear to what extent older people and their health professionals consciously consider the values they hold in adopting and recommending use of health-supporting SST in older people.

This study was undertaken to identify the factors involved in older people’s adoption and satisfaction with health-related SSTs, from the perspective of people 70 years and over and health professionals with oversight of SSTs in older people. While older age is generally considered to occur at about age 65, there is considerable variability in lifespan in developed and developing countries [28]. In Australia where the data were collected, the average lifespan is currently 86 years for women and 82 years for men, with retirement age being 67 years and expected to rise to 69 years by 2025 [29]. For this study it was, therefore, determined that the older population would be identified as people 70 years and over, which is the age when major health issues start to surface for most Australians [29]. The SSTs of interest in this study included any internet-based technological devices/applications designed to assist older people to monitor and manage their health, well-being and function. SST examples included but were not limited to health apps, personal response systems, medical alarm systems and wearable sensors. SSTs that were not of interest included sophisticated healthcare technologies requiring external setup/intervention, and products that are not internet-based.

The following research questions were posed:(1)What SSTs are useful to people 70 years and above in maintaining their health?(2)What factors are involved in the adoption of SSTs by people 70 years and over?(3)What are the benefits and issues for people 70 years and over in using SSTs?

## 2. Materials and Methods

### 2.1. Design

This prospective, mixed-methods study was undertaken from February 2020 to October 2021, using quantitative surveys and individual semi-structured interviews.

### 2.2. Ethics

The study protocol, measures and advertisement, and the Participant Information Statements and Consent Forms (PISCFs) were approved by the University of New South Wales Human Research Ethics Committee (HC200402).

### 2.3. Participants

There were two population groups: people 70 years and over who used self-supporting health technologies (STT) and healthcare professionals who had oversight of self-supporting health technologies (STT) in older people.

A purposive sampling approach was utilised to recruit research participants who lived in the Sydney metropolitan area and had experience of SST use in older age. The target survey population included (n = 25) people 70 years and over and (n = 25) healthcare professionals. The interview participant number was determined by data saturation. As the SSTs of interest in this study covered a wide range of services available to older people, both participant groups were required to have knowledge and/or experience with these applications. Older people were eligible to participate if they were 70 years of age or older; used electronic media; were currently using self-servicing health technology (SST); were living independently in the community; were able to read, write and communicate in English to a comfortable level; had hearing and speech that was sufficient to engage in a verbal interview in English; and able to provide informed consent. People 70 years and over who were unlikely to be able to understand and provide answers to survey and interview questions in English, and who were unable to provide their own informed consent, were not eligible. Health professionals who met the following criteria were eligible to participate in the study: providing oversight of self-supporting health technologies (STTs) in people 70 years and above; used electronic media; were able to read, write and communicate in English to a comfortable level; had hearing and speech sufficient to engage in a verbal interview in English; and were able to provide informed consent. Health professionals invited included physicians, nurses, chiropractors, physiotherapists and geriatricians who routinely provided healthcare services to older people. Health professionals were ineligible for the study if they had no knowledge or experience with SST use in older age and were not providing services to older people.

### 2.4. Participant Recruitment and Consent

People 70 years and above were recruited through organisations and services that agreed to allow distribution of the study advertisement to their members. These organisations/services included the senior’s group Older Women’s Network (OWN), and retirement villages, churches, Meals on Wheels and GP practices. The on-line study advertisement was provided to all older members of these organisations by administrative staff (or nominee as approved by centre/service executive) and it was made available to all members through organisational newsletters (online and hard copy). Health professionals who met the inclusion criteria were recruited through primary healthcare services (GP practices and community support/health services) that agreed to allow distribution of the study advertisement to their staff by the office administrator, and through regular service communication routes (online and hard copy).

Interested persons were asked to contact the lead researcher (L.L.) directly and nominate themselves to join the study via email or telephone. After providing detailed verbal information about the study, researchers (L.L. and L.C.) provided the relevant PISCF to interested persons via email, giving them at least one week to provide written consent to complete the on-line survey to indicate their willingness to participate in a one-on-one telephone interview. For those who also agreed to participate in the interview, L.L. and L.C. provided further details of the study aims, procedures and intended outcomes prior to obtaining their consent, including permission to audio-record the interviews. Participants were made aware that they were free to withdraw from the study at any time with no penalties. Time was given for participants to ask and clarify any questions regarding the study before being asked to sign the PISCF. Participants were given a signed copy of the PISCF. Electronic copies of the PISCF were stored in a password-protected computer system on the secure university server.

### 2.5. Researchers

The researchers included two experienced female PhD qualified researchers (L.L. and L.C.) and a male PhD student (Y.G.Z.). The research team has considerable experience in conducting both qualitative and quantitative research in information technology and innovative technologies and their applications in society, cybersecurity, ageing transitions, chronic illness prevention and management, older person healthcare models and aged care policy. Apart from L.L.’s relationship with two health professionals, the researchers had no previous relationship with other study participants. This conflict of interest was addressed by having L.C. undertake the two health professional interviews.

### 2.6. Measurement

Study measures included a survey and one-on-one interviews. To answer the three research questions (above), the survey and interview questions were guided by the Customer-Dominant Logic (CDL) framework [30], which focuses on service/product value from the perspectives of value-in use, the consumer’s own context and the consumer’s experience of service. In CDL, “value” refers to a relative preference evaluation concerning a particular service or product, e.g., blood pressure monitor. “Value-in-use” refers to the value which emerges only when the individual uses the service or product. “Use” is defined narrowly, e.g., the service or product might be used before purchase, during interaction or after purchase [30]. The draft survey and interview questions were piloted with one healthcare professional and one person 70 years and over. These individuals provided verbal and written advice on the focus, structure and wording of the questions, which informed finalisation, e.g., clarification of concepts with examples, and question wording, order and process.

Surveys—Guided by CDL [30], the 48-item survey components included: participant demographics, and ratings of the selected technology according to: SERQ (service quality)—17 items); AwareHP (awareness of functionality of SST—7 items); RMAP (important characteristics of selected SSTs—9 items); PAEM (relevancy of SST for older people—4 items) and DP (level of agreement to the DP statements—7 items for persons 70 and above, 6 items for health professionals). Respondents were requested to complete all survey questions. Surveys are available on request to the authors. Researcher L.L. took responsibility for survey distribution and receipt. The survey was administered online to both participant groups using Qualtrics (n = 20) or posted to participants upon request in a paper-based form (n = 2). Participants were requested to answer closed questions by selecting the options that were most appropriate for them. All people consented completed the online survey via Qualtrics, which captured data anonymously in digital form, or on paper-based surveys which were manually transcribed by L.L.

Interviews—The CDL [30] provided the framework for the 12 interview questions, which focused on the following constructs: What kind of SSTs are older adults engaged in and why? In what types of SST use does value emerge for the older person? What are the dimensions and drivers of SST experience? Which are the boundaries of SST use? What factors in different contexts are critical for favourable and unfavourable value-in-use of SSTs? How, when and where can value-in-use of SSTs be measured? How can value-in-use be explicitly expressed? How can older persons’ experiences with SST be better supported? Five (n = 5) survey respondents 70 years and over and two (n = 2) health professional respondents agreed to participate in the one-on-one semi-structured interviews. The 60–90-min interviews were conducted by researchers L.L. and L.C. using Skype and Zoom, or by telephone, depending on participant preference. Participant responses were audio-recorded and transcribed verbatim. Field notes were not recorded. Interview participants were posted a gift voucher at the conclusion of the interview and were provided with summary findings.

### 2.7. Data Handling, Entry, Storage

Researchers L.L. and L.C. transcribed each of the interviews they conducted within 24 h. Transcript accuracy was checked by listening to and cross checking the transcriptions with the audio recordings and comparing them with the hand-written notes obtained during interviews. Inadvertent recording of the participant names and other identifying details, such as workplace, were erased from the audio recordings and replaced with data codes in the transcribed data; in order to ensure study site and participant confidentiality, participants were allocated unique identifier codes, e.g., P01 (Older Person, interview 1), HP01 (Health Professional, interview 1). Electronic copies of the audio-recordings and interview transcripts were stored securely in Microsoft OneDrive offered by the university, using two-factor authentication with the Microsoft Authenticator app. Data files were encrypted on Microsoft servers, and an encrypted network connection was used to transfer files. A password generator was used to generate strong unique passwords for access to OneDrive by researchers L.L., L.C. and Y.G.Z. As approved by the university’s Human Research Ethics Committee, all study materials including computer data files and records (e.g., PISCFs and interview data) will be retained in the university’s data archive OneDrive for 5 years after publication of the research findings, and will then be destroyed by shredding of paper files and erasure of computer files according to our university policy (see Section 2.2 Ethics).

### 2.8. Data Analysis

#### 2.8.1. Quantitative Data

Demographic and survey data were entered into SPSS [31], and descriptive statistics were conducted, including frequency, mean, standard deviation (SD), median, range and minimum and maximum scores wherever relevant. Analyses on survey constructs SERQ (16 items), RMAP (9 items), PAEM (4 items) and DP (7 items for persons 70 and above, 6 items for HP) were conducted by using a one sample *t*-test at a significance level of α = 0.05 to examine the statistically significant agreements of each statement in each component. Independent samples *t*-test at 95% confidence intervals was employed to compare the scores in SERQ (16 items), RMAP (9 items), PAEM (4 items) and DP (5 items are comparable). Given the small sample size, Levene’s test [32] was used to calculate equality of variance at α = 0.05, determining whether equal variance was assumed before the corresponding t-value, degree of freedom (df) and *p*-value were calculated.

#### 2.8.2. Qualitative Data

Data saturation for all questions was achieved by the seventh interview. An inductive thematic analysis was undertaken with the interview data [33], as follows. Data familiarisation was conducted independently by L.L. and L.C., who read through the interview transcripts to get an overview of the depth, richness and diversity of the data collected. These data were inspected independently by L.L. and L.C., who tagged participant quotes that directly responded to the a-priori interview questions. Data codes arising from participant quotes were independently developed by L.L. and L.C., and the codes were entered to data tables under each of the interview questions. The data codes and supporting participant quotes were then tabulated separately for participants 70 years and over and health professional participants. Initial inter-rater reliability (IRR) of data codes independently allocated by L.L. and L.C. was calculated as the number of agreed codes over the total number of codes allocated, which achieved 80% agreement [34]. L.L. and L.C. then compared the data codes allocated to both participant populations and agreed on the final naming and allocation of the data codes. The common themes arising from the agreed codes were identified and named by L.L. and L.C. through close inspection and discussion of the participant quotes which supported the agreed data codes [35]. These study methods adhered to the COnsolidated Criteria for REporting Qualitative research (COREQ) [36].

## 3. Results

### 3.1. Participant Characteristics

#### 3.1.1. Survey Respondents

Of a possible 50 people invited to complete the survey, 22 individuals self-identified as meeting the relevant eligibility criteria and consented (44%).

People 70 years and over (n = 12): Eight of twelve participants were female, nine were of Anglo Australian (n = 9, 75%) ethnicity and all were over the age of 70, with half (n = 6) in the 70–74 age range. Eleven participants were retirees who lived in private homes with their partners. Their education levels were evenly split between undergraduate and postgraduate qualifications. Their most prevalent chronic health condition was cancer (n = 4, 33%) followed by heart disease (n = 3, 25%), thyroid condition (n = 3, 25%) and a neurological condition (n = 2, 17%). Health-related SSTs used included an electronic blood pressure monitor (n = 5), online health resource apps (n = 5), blood glucose monitor (n = 3), Dexcom clarity diabetes APP (n = 2), fit-bit/smart watches (n = 3) and a hearing aid (n = 1).

Health professionals (n = 10): Half (n = 5) of the health professionals were general practitioners/geriatricians/specialist medical practitioners whilst others were nurses (n = 2) and allied health practitioners (chiropractor, physiotherapist) (n = 3), composed of six females and four males. All had tertiary qualifications, with ages ranging from 31 to 40 (n = 3), 41 to 50 (n = 3) and 51 to 65 (n = 4). The most common SST-supported health conditions of their older patients include musculoskeletal (n = 6) and neurological conditions (n = 5), followed by mental health (n = 4) and cardiac conditions (n = 3).

#### 3.1.2. Interviewees

People 70 years and over (n = 5): two interviewees were male and three were female. The age of one female was in the range 82–91. All other interviewees were in the age range 71–81. They used many different SSTs for different purposes such as blood glucose monitoring, hearing aid, fitness monitoring, wellbeing and fitness tracking, nutrition and medication tracking, emotional/brain health, COVID-19 tracking and fall detection.

Health professionals (n = 2): one interviewee was a female general practitioner (age range 51–61), the other was a male chiropractor (age range 31–50).

### 3.2. Research Questions

#### 3.2.1. Research Question 1: What SSTs Are Useful to People 70 Years and Above in Maintaining their Health?

Both Survey and Interview Data Provided Answers to this Research Question

##### Survey Responses

Owing to the small sample size, the results are presented descriptively as means; *p*-values have been presented where suitable [32]. Participants 70 years and over (n = 12) gave a rating of between 3 and 4 (medium-high) on a five-point scale (Extremely Low = 1, Low = 2, Medium = 3, High = 4, Extremely High = 5) on their literacy and confidence in using IT. On average, they used two types of SSTs for health maintenance, with most using a smartphone for between 6 and 12 years. Their main reasons for using SSTs were to improve health literacy, participate in healthcare decisions, communicate with health professionals and take control of their health (*p* < 0.001). They were satisfied with the services provided by the healthcare technologies and agreed that SSTs were beneficial (*p* = 0.003), did not fear using them (*p* < 0.05) and would recommend persons of same age range to try out SSTs to support their health (*p* = 0.003). Participants gave a rating of between 4 and 5 (out of 5) for SST most beneficial outcomes of health literacy (mean 4.58), communicating with the health professionals (mean 4.42), participating in healthcare decisions (mean 4.42) and taking control of their health (mean 4.42).

On a five-point scale (Extremely Low = 1, Low = 2, Medium = 3, High = 4, Extremely High = 5), health professionals rated their literacy and confidence in using IT between 3 and 4 (medium-high) and had owned a smartphone for between 5 and 11 years. Health professionals indicated that they were aware of SST functionality in health maintenance for older people (rating 4.5, *p* < 0.05) but were dissatisfied with many current SSTs and services available to older people (*p* = 1.000). They were generally ambivalent (rating of 3 out of 5) about the older person’s reliance on SST in health maintenance, preferring remote monitoring by a health professional (rating 4.5, *p* > 0.05). The most favoured SST applications for older people (rating 3–4) included health literacy (mean 3.33), communication with the health professional (mean 3.34), participating in healthcare decisions (mean 3.56), taking control of their health (mean 3.33) and safety alert functionality (mean 3.4).

##### Interview Responses

One common interview theme emerged in answering Research Question 1:

Theme: Willingness to adopt and to recommend SST applications supporting older people’s health.

Participants 70 years and over are denoted by their unique identifier P01–P05, and health professionals are denoted by their unique identifier HP01 or HP02. Health professionals recognised and approved of the broad diversity of health-related SSTs available to older people. The traditional SSTs being used by participants 70 years and older included mainly hearing aids (P01), blood glucose monitor (P03, P04), and pacemaker (P03). Health professional participants were aware of and recommended personal alarms (HP01), mobile phones (HP01, HP02), electronic clocks (HP01) and advanced technology hearing aids (HP01, HP02). Numerous innovative technologies were being applied in participant health maintenance, such as wearable WIFI transmission device (P01), Bluetooth transmitter (P01), smartphone apps (P02, HP01), smartwatches (P03, P04) and smartphones (P03, P04, P05, HP02). Participants identified that innovative SSTs were becoming more widely recognised, including “…with the fitness app (on my smartphone) that lets me count every day how many steps that I was doing” (P02), and “…I have an app on my phone that I can look and see for the last seven days what my average blood glucose readings are. I can see whether it is better than last time or not as good as…” (P03). Many participants also had previous experiences with or were willing to try out new devices such as smartwatches. This was apparent through P04′s plan to acquire a smartwatch: “I am getting a Galaxy watch, which is a new product from Samsung…it has got the added function of fall detection and alarm. It is like a watch so you do not wear it on your neck” (P04).

#### 3.2.2. Research Question 2: What Factors Are Involved in the Adoption of SSTs by People 70 Years and Over?

Research Question 2 was answered by survey and interview responses.

##### Survey Responses

On a 5-point rating scale (Strongly Disagree = 1, Disagree = 2, Neither Agree nor Disagree = 3, Agree = 4, Strongly Agree = 5), participants 70 years and older rated the most important factors in SST adoption as being affordable, safe, controllable, protecting privacy, allowing choice of use, relevant to needs, user-friendly, reliable and allowing for integration (*p* < 0.001). Other favourable features included being up-to-date, appealing, accurate, dependable, trustworthy, secure and reacting promptly (*p* < 0.05). Values such as trust, confidence and faith in the SST; meeting their health needs; and gaining insights on their health condition/s (*p* < 0.05) were desirable. Factors such as SST visibility, sensitivity, immediate alerts, personalisation and privacy protection held less value for the participants (*p* = 0.053–*p* = 0.082).

Health professionals gave a mean rating of 4–5 (out of 5) for the following most important SST adoption factors in regard to older people’s health maintenance: reliable, accurate, dependable, trustworthy, reacts promptly, immediate alert, controllable, user-friendly and relevant to the health condition (*p* < 0.05, all means > 3). Privacy and choice of use received lower ratings (mean rating 3–4).

##### Interview Responses

Two common themes and four associated sub-themes emerged in answering Research Question 2: Theme (1) Adoption of health-related SST prior to older age, and Theme (2) Knowledge gaps in SST health support. Adoption of health-supporting SST prior to older age was factor in continued use as the older person’s health deteriorated. Most of the older participants, as well as the health professionals, had limited understanding of more advanced health-related SSTs; knowledge was restricted to applications they experience with. Limited understanding of how SST could be used to support older people’s health was mainly due to difficulty in obtaining accurate and reliable information on their appropriateness. Participants advised that there needs to be more readily available and accessible information and education available to older people and health professionals to increase their adoption in health monitoring and maintenance (Table 1).

#### 3.2.3. Research Question 3: What Are the Benefits and Issues for People 70 Years and Over in Using Health-Related SSTs?

This question was answered through interview responses alone.

Three common themes and nine associated sub-themes emerged: Theme (1) Personal values influence perceived benefits of SST in health maintenance, Theme (2) Constraints to SST adoption and Theme (3) Strategies for better SST adoption and use.

All participants identified the importance of SSTs in managing chronic physical and mental illness, including those which facilitated online or remote health assessments and clear communication with health practitioners. Older participants also highly valued the role of SSTs in making lifestyle choices that helped in taking personal control of their health. Value was also placed on SST integration with existing treatments, health protocols and health infrastructure, which empowered the older person to make health decisions in consultation with their health practitioners. Common challenges to the adoption of health supporting SST by older people were technology phobia and low technological knowledge and skills. More sophisticated health SSTs such as smart phones, GPS devices, online self-assessment tools and smart watches were considered inaccessible to them, especially for the very old and those with significant cognitive impairment. Participants emphasised that this could be redressed by making SST more accessible, user-friendly and affordable for older people (Table 2).

## 4. Discussion

Study participants 70 years and over showed a willingness to use STTs, despite access and support issues, and complexity of some SST technological features. They would be willing to adopt more sophisticated SSTs if there was better education and training prior to adoption, more readily available support once adopted and more user-friendly design features. Consequently, there is a need for investment in user-friendly technology and support services, as well as engagement of healthcare professionals in promoting health-supporting SSTs for older people.

The survey responses indicate that participants 70 years and over expressed more positive user experience with SSTs for health maintenance and were more satisfied (mean rating > 4) with the SST services provided, compared with health professional ratings (mean rating 3) of older people’s SST usage patterns. Participants 70 years and over also had greater recognition of the inherent benefits of SSTs in health maintenance (mean rating > 4) than did health professionals, who considered that SSTs were challenging for older people (mean rating 4) and more training was needed in the use of SSTs (mean rating 4). A clear distinction between responses occurred regarding the benefits of SSTs in health maintenance, however, where participants 70 years and over were more concerned about improving their health literacy with SSTs, while health professionals identified the value of SSTs in helping the older person to participate in healthcare decisions.

There were few differences in ratings given by people 70 years and over and health professionals on the important characteristics of health-supporting SSTs for older people. Small differences in ratings included the relevance of SST safety, affordability, choice of use and privacy protection, with people 70 years and over rating these characteristics as slightly more important compared with health professionals. The discrepancy between older people’s and health professionals’ perceptions could be due to the different types of SSTs being considered, the different application scenarios, the variation in respondents’ SST exposure and experience and the different levels of competence in their use.

While there were only seven interviewees, their responses reinforced and helped to explain the survey results. Interviewees recognised the diversity of SSTs in health maintenance, considering that many of them were suitable for use in older people. There was common agreement that improved self-management of chronic health conditions can be achieved through SST monitoring and decision-making capabilities, enhanced communication between the older person and their healthcare practitioner and in helping individuals monitor and manage their personal health and wellbeing. These finding are in agreement with results of systematic and narrative literature reviews on SST benefits [4,5,15].

Knowledge of SST availability and functionality was, however, confined to SSTs that participants had experience with. Except for those people 70 years and over who were early adopters of technology, most older participants tended make conservative choices in their SST selections. In agreement with systematic literature reviews of experimental and mixed methods SST studies [6,9,17], other reasons for more conservative exploration of health-related SSTs included a degree of technology phobia; SST complexity; difficulty with navigating the SST hardware and software; and having health practitioners who do not actively recommend more advanced SSTs due to their own lack of knowledge, confidence, skill and scepticism of novel SST functions [5].

As proposed with the CDL model [30], value-based motivation can explain the older person’s adoption of SSTs, for example, their use in self-assessment of bowel health [23]. Functional, emotional and social values played a role in the adoption of SSTs in relation to its perceived advantage and ease of use. Internal factors such as technology compatibility, trialability and perceived image were not as important. The values-based consideration of SST adoption in older participants supports application of a framework comprising the technology acceptance model, innovation diffusion theory, technological innovativeness and protection motivation theory, applicable equally to older SST users surveyed in developing and developed countries [37]. Evaluation of 426 surveys (222 females, 204 males) using a partial least squares structural equation model with data grouped by gender confirmed the framework’s proposition that perceived advantage and ease of use have a significant effect (*p* = 0.001) on intention of older people to adopt SSTs [27].

In the current study, perceived advantage included how well SST functioned, i.e., being able to be controlled, offering choice, safe, user-friendly, reliable, affordable, involving integration into everyday life and with traditional healthcare support. These results concur with surveys (N = 1095) of older Finns on the role of functional value in SST adoption [38]. The emotional values of trust, confidence and faith in SST function were also found to be important influencers. As identified in systematic reviews of barriers to SST use [39], older people were willing to embrace user-friendly SST if they were perceived to improve personal safety, particularly in relation to real time monitoring and connection to response systems. In contrast to the Finnish survey results [38], older study participants were not concerned with privacy of the SST-captured data sharing with their health practitioner and family members. In agreement with the findings of qualitative literature reviews [6,10], participants considered that health-supporting SSTs must be secure, fluent and relevant to the older person’s day-to-day life to motivate them to share this type of information with others, including for people living with dementia [40].

The social value of SST [30] for some older participants was in having greater control of their health condition when in public situations. This included being able to monitor and manipulate their heart rate and rhythm via pacemaker and activity adaptation, improve reception with hearing aids, monitor and adjust blood glucose and insulin levels, undertake continuous risk assessment in fall prevention and teach peers how to use smartphones and watches for health support. Online surveys of SST use in Australians 65 years and over (N = 737) during the COVID-19 pandemic concur with these sentiments [41].

Confirming the CDL [30] influence in older people’s SST use, the participants’ positive experiences helped incentivise further adoption, while negative experiences resulted in the phenomenon of technology phobia. This was a concern for health professional survey respondents, some of whom believed that older people cannot cope with modern technology. Focus groups with older people of different cultures on SST desirability also placed a large amount of emphasis on usability [8,16] and accessibility [39]. The results indicate that making SSTs more user-friendly can increase adoption by older people, since slow adoption of more complex SSTs was mainly due to usability issues. Accessibility barriers commonly raised amongst older participants included lack of knowledge and technological support for suitable technologies. These findings concur with focus group responses of older Chinese individuals (N = 24) [8], emphasising the importance of digital literacy for both older people and the health professionals who support them, and of better access to technology knowledge and education for both.

Conflicted survey responses of SSTs both supporting and hampering health maintenance function is a novel finding. Older people were more optimistic in recognising the health benefits of SSTs than were health professionals, as identified through interviews with 59 dyads of older people and their carers in the Netherlands [42]. Older participants viewed the technology as a supporting tool that enabled them to maintain autonomy and self-care, thus providing greater ability to take control of their health, notwithstanding their acknowledgement of the difficulties facing people with a cognitive impairment in using such technology, as previously identified [26]. Health professionals predominantly viewed SSTs as hampering traditional health maintenance support, suggesting that older people cannot cope with more complex SSTs and that human judgement regarding health status may be more reliable [43]. In recommending SST, therefore, most health professionals were in favour of more traditional applications such as health monitoring and tracking for older people. The predominance of this view was identified by older participants as giving rise to many missed opportunities to apply SSTs in preventative health initiatives, including assisting people with sensory and/or cognitive impairment to prevent further deterioration in function [17,20]. Many of the health professionals surveyed were cautious about exploring these potential health-supporting SSTs for older people.

To redress unwarranted caution, health professionals need to become more knowledgeable about SSTs developed specifically to support commonly occurring health issues in older age, including their functions, cost, access, technology support services and utility. Armed with this knowledge, health professionals will be more able to provide older people with relevant information about suitable SSTs for specific health issues, advice on SST access and support services and ways in which SSTs can be used to complement traditional health support [4,17]. Younger family members also have a role to play in encouraging SST use for health prevention and support in older people, for example, with fitness/heart monitoring and sensory changes. Since younger family members are more likely to become aware of changes in the older person’s usual health status such as in vision, hearing, agility, strength, endurance, gait, cognition and mood, they can encourage and assist the older person to make use of suitable SSTs in maintaining specific health functions [40,44]. Holding joint consultations with the older person and younger family members regarding potential use of SSTs in managing chronic illness, such as high blood pressure or diabetes, would also encourage and support the older person and health professionals to make better use of SSTs as a complement to traditional healthcare.

Areas for further investigation include ways of promoting and educating health professionals on the role that SST can play a more holistic role in older people’s healthcare. For example, education might include exploring the concepts that embracing SST in older age is not only linked to the immediate process and outcomes, but extends beyond to the value added to CDL, which is mainly associated with personal goals, e.g., lifestyle, health or social engagement [27]. Such experiences confirm that value-in-context is dynamic because experiences are continuously accumulated with continued use. Further research is also required to examine whether increased SST use by older people can be achieved through public education programs, devising resources required by consumers in self-services such as screening kits that are easy to use and producing instructional materials that are interactive and easy to understand and follow. It is especially important for older people with a cognitive and/or sensory impairment for SSTs to be “humanised” in their features, so that instructions/prompts for use are accessible, understandable, uncomplicated and personalised [21,28,38].

### Strengths and Limitations

The study’s mixed methods, which were informed by the CDL framework [30], enabled a conceptual analysis of SST desirability and adaptations required for promoting their use in older people health. Viewed from the perspective of people 70 years and over and health professionals who regularly provided healthcare support to older people, the findings provide a reference lens for future investigation of value-based SST utility and usability in older age health management. The participants’ knowledge, experiences and approaches to managing health issues with SSTs applications have resulted in an information-rich sample which correlates with the research questions posed [33].

Augmentation of the survey data with participant insights obtained from in-depth interviews is a study strength. Scientific rigor occurred in this qualitative component of the study through application of the following principles: confirmability, dependability and credibility [34]. During data collection, one-on-one discussions with study participants ensured credibility, while during data analysis, peer debriefing and triangulation enhanced the understanding of variations of participant experiences within the study sites (dependability). Regular review and reflections by the investigators on survey response grouping and interview transcript coding was also employed to achieve confirmability.

A study strength included the participation of researchers with diverse backgrounds appropriate for this research topic, including information technology/systems design, implementation, and consulting experience; electronic healthcare knowledge; and gerontological expertise. This collaboration was particularly fruitful when analysing and interpreting survey and interview responses, following an agreed and consistent approach and achieving investigator triangulation [34,35]. Reflexology was achieved by attending systematically to the context of knowledge construction, especially to the effect of the researchers, at every step of the research process [35]. Additionally, the authors’ personal and professional experiences with older adult healthcare and gerontological knowledge were subject to considerable reflection when devising the research questions, making decisions on how these questions might be posed and probed at interview, and when reflecting on participant responses [33].

The study limitations include the small sample size, purposive sampling of people with access to online media and with experience of SST use in older people [44] and knowledge and ability to answer focused questions [33]. Owing to the small sample size, it was not feasible to undertake thematic analysis by age group [45]. While recruiting people 70 years and over via established organisations, communities and networks of aged populations in Sydney, including retirement villages, supported aged care services and churches, low participant uptake proved disappointing. Low recruitment could possibly relate to the participant inclusion and exclusion criteria of requiring direct experience with health supporting SST and electronic media. The results confirmed that some older people would not use SSTs because of technological phobia, low knowledge and skills with technology and difficulties faced with technological complexity, and poor level support from tech providers; such people would not be interested in participating. The study was undertaken during the COVID-19 pandemic which also impacted access to potentially suitable older people and health professionals, given the requirement to adhere to the approved recruitment protocol, which necessitated online provision of English language study information. Thus, a study limitation and potential study bias was the necessity to include only English speakers, which likely prevented older people from culturally diverse language groups to contribute insights on SST use. Other limitations included not assessing the participants’ skills in SST use, which may have confounded the results [37], and the exclusion of people with a high-level of cognitive impairment, some of whom may have had valuable insights of health-supporting SST features they are able to navigate and manipulate. Further studies could redress these limitations by exploring the relationships between technological skill, cultural background and cognitive status, which may have a bearing on SST adoption, functionality and other benefits for older people with specific needs, as well as barriers to use.

## 5. Implications

Since CDL [30] has been shown to play a role in health-related SST adoption by older people, human behaviour and preferences need to be taken into account in the design of SSTs. It would be beneficial for older people, their family members and health professionals to be involved in health-related SST design, since user-compatible functionality and user-friendly design interfaces will often influence the adoption and effectiveness of the technology more than the nature of the technology itself. Technological challenges with using modern SSTs may be overcome by support and guidance from younger family members and through technical support providers, especially in the initial setup stage.

Health professionals can play an important role in recommending SST adoption and oversight of STT use in older peoples’ health support. To do so, they must have awareness and knowledge of the availability and characteristics/features of SSTs available in the market. Health professionals who are “digital natives” (usually born after 1980s) [46] are comfortable in the digital age and would be more inclined to learn about SSTs and recommend them. Health professionals who are “digital immigrants” (born before 1980s), however, may be less inclined to recommend SSTs to the older generation as they themselves are less comfortable with technologies compared to the digital natives [46]. It would be beneficial for older health professionals to be educated/trained in SSTs suitability for promoting their use in older people.

Accessibility issues, including technological complexity, affordability and education/training support must be addressed to increase SST uptake in older people. Well-designed, user-friendly SSTs which include in-built education/training tailored to individual knowledge deficits are also needed to meet a variety of older people’s health issues, including both the functional and non-functional (e.g., reliability, accuracy) requirements [39]. Further value-based market research with older people is needed to identify their particular needs, desires and issues with SSTs designed to support health. Collaborative interdisciplinary projects involving affected stakeholders, e.g., technology (including user experience and user interface) designers and developers, older people, family members, health professionals and health policy makers, will assist in building and integrating SSTs into the healthcare and lifestyle of older people.

## 6. Conclusions

Since participants identified that many SST functions are too complex for older people, the design of health-supporting SSTs must include advice from health professionals, younger family members and others who are aware of the older person’s technological capabilities. User friendly SST design can be achieved when technology experts and user groups collaborate in their design and testing. Older people need far more practical support for SST setup, ongoing use, installing upgrades and undertaking maintenance than is currently available; this would also be likely for those SSTs which require health professional oversight. Without access to user-friendly technological support, it is unlikely that health professionals will recommend SST use in older people, especially for those with cognitive changes. Greater uptake of SST in supporting older person health, therefore, requires targeted SST education for older people, family members and health professionals, making SSTs more affordable, accessible and user-friendly for older people, and providing more accessible technological support to users who struggle with technological complexity. A possible solution to increasing health professional support of SST use in older person health is for technology companies to lease suitable SSTs to health professionals so that they can oversee trial-runs with older people who might benefit from their use in specific areas of health assessment and management. Ready access to technology support services and upgrades would be necessary for this to be taken up by health professionals in providing oversight of the older person’s health.

## Figures and Tables

**Table 1 healthcare-10-00738-t001:** Research Question 2 themes.

Theme	Sub-Theme	Participant Quotes
1. Adoption of health-related SST prior to older age.	1.1 To manage deteriorating health.	“…for hearing loss” (P01, HP01). “…improve fitness” (P03, P04, HP02). “…mental health…and senses (HP02) “…my well-being” (P05).
	1.2 Early adoption helps to maintain health and function.	“…videos, audios … Zoom meetings, self-help books… on-line events, learning platforms…increases health knowledge…I can’t express enough how it has helped my mental well-being” (P05). “…very good to increase … knowledge and … understanding about an issue with health” (P02). “…I read what they say and I take note … I would not be alive without (this) medical technology” (P03). “… helped her to contact her daughter and let her know where she was lost…” (HP02)
2. Knowledge gaps in SST health support.	2.1 Limitations in understanding SST functionality.	“I don’t think there’s any technology … that would give early warning signs of a developing problem.…I would definitely use something if there was…” (P01). “… I have not used that one …don’t know about it.” (P02). “… no, only use ones I have experience in” (P04). “I only recommend SST health monitoring that is low-key … like glucometers…sphygmomanometers, the electronic one that I know about.” (HP01).
	2.2 Difficulty in obtaining reliable information about SSTs	“… tried to find out…I rang my doctor’s surgery… never had a clue of what I was talking about.” (P01). “…the GP at my generation, they are even scared of using computers…you have to ask your son or daughter” (P02). “… you have to read all you can about it and see whether you think it will help you… on the internet” (P03). ” …I don’t…rely on technology to make the diagnosis” (HP01). “…there needs to be more readily available …information and education …. for older people (about) SSTs” (HP02).

**Table 2 healthcare-10-00738-t002:** Research Question 3 themes.

Theme	Sub-Theme	Participant Quotes
1.Personal values influence perceived benefits of health-related SST.	1.1 SSTs are important in managing chronic illness.	“ …yes, for hearing (P01, HP01), “heart condition … and diabetes” (P04, HP01). “…emotional health” (P05, HP02). “I have a pacemaker …it is (automatically) reading my heart every night and sending messages” (P03) “…when patients have been able to work their phone, they have sent me some pictures of things…” (HP01). “… assess, critique what we will be doing and advise on areas where they can improve” (HP02).
	1.2 SSTs enhance autonomy and lifestyle choices.	“…to be able to take control of my own lifestyle is of major importance” (P05). “…the knowledge of nutrition… to be aware what the calories are and what is carbohydrates” (P06). “I had a patient that got lost driving. She had the mental capacity to turn into a carpark and she didn’t know where she was, so she activated the alarm and she told me within minutes that her daughter spoke to her on the phone …” (HP01). “… it helps them to take …ownership and responsibility for their lifestyle and health.to know what’s required… also helps them to set targets and the outcome measures they can work towards” (HP02).
	1.3 SSTs enable integration with existing health procedures.	“(data on blood glucose level) is collected and sorted. When you go to the diabetes specialist, the endocrinologist, he is going to attach it to a reader…” (P04). “There are also some (cardiac) fault-detection devices…they need this” (P01).: “… one patient just over seventy… got a Fitbit watch and she was showing me… that she actually did have four, five hours of proximal SVT nearly every day …occurring even in her sleep…so I convinced her that she should see a cardiologist… and it pushed her to have an ablative procedure done…” (HP01). “… helping them to participate in health care decisions… you can’t make an informed decision unless you have access to the facts” (HP02).
2. Constraints to SST adoption.	2.1 Technological phobia is a barrier to SST adoption.	“… not many friends want to accept it because they are afraid…of it…. older people are usually scared of technology… they are not very good at technology…” (P02). “… it really worries you a bit if things go wrong” (P03), “…you need to have some sort of basic understanding of the technologies so that they can feel more comfortable when you want to use it” (P04). “…they cannot remember the password and panic …and then get terrified of it…” (HP02).
	2.2 Technology complexity poses challenges in SST use.	“There’s a limit to what I can do” (P01), “…depends very much on … what condition the person is in” (P04). “…people with dementia…they would not know how to use it” (P03). “… a lot of my patients are not very good…. especially in the over eighty age group, to know to be able to use mobile phones even…. they just can’t use that technology.” (HP01). “…the biggest challenge … is their inability to log in to the system…and… to work out the trail … the technology can be a bit confusing … and navigating the different menus would be too difficult for them, too complex” (HP02).
	2.3 Inaccessibility of SSTs for older people	“…could not manage it… even with the smartphone …because they do not have the basic knowledge… so they get stuck” (P02). “…that’s the problem, the smaller the device it makes it more difficult” (P04).“… my patients don’t have computers… even the e-health record … they don’t go up there and check it…our patients can’t do that.” (HP01). “…there are definitely some limitations with the very oldest, such as their self-assessment of some aspects of their lifestyle” (HP02). “… a smart watch, an apple watch or something like that… it’s maybe too fiddly and difficult to access the different programs…” (HP02).
3. Strategies for better SST adoption and use	3.1 Make technology user friendly for older people.	“…it would be better if there were more directions (embedded) in the technology…” (P01). “It should be easy for the user, particularly the user at my generation… it needs to be something quite simple and easy to understand…easy to get access into it…know how to use it…know how to search and how to find what I want to find…” (P02). “…make it easy … you just press the top and go one, two, three, and it talks to you, it gives you information….” (P03). “… very important that self-service technology information helps them to participate in health care decisions… to help the person to have control over their health…it kind of helps as an accountability partner and to motivate them to do some of the things that are recommended.” (HP02).
	3.2 Increase SST accessibility for older people.	“…electronic design, obviously, I need … help on that.” (P01). “…blood glucose monitoring device …if your unit is not working, well…” (P04). “…it can be user-friendly if you study it up and look at it…but someone has to show you.” (P03). “…need more access to videos, audios… on-line events, learning platforms…” (P05). “…larger keys/screens…speed dials…automated functions …education…” (HP01), “…the easier it is to use the technology, the more compelling it is that the person will be willing to use it.” (HP02).
	3.3 Make SSTs affordable for older people.	“ …will have to wait ‘til next year… when they (more advanced technology aids) become available… because they are very expensive…” (P01) “… the purchase price is pretty stiff… seven hundred, maybe eighty dollars… and the monitoring fee is fifty dollars per month.” (P04). “…it costs me $700 for the three months. It is not cheap and a lot of people just cannot afford it.” (P03). “….it has to be not very expensive, or it could be subsidised…” (HP01).

## Data Availability

The data presented in this study are available by request to the lead author.

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
