# Peer review of "Exploring Adoption and Satisfaction with Self-Service Health Technology in Older Age: Perspectives of Healthcare Professionals and Older People"

_healthcare, 2022, doi:10.3390/healthcare10040738_

Round 1

Reviewer 1 Report

Dear authors,

please find my review to your manuscript attached in pdf.

Best regards!

Author Response

The rejoinder uploaded addresses all requested amendments 

Reviewer 2 Report

This is an interesting study. The emerging findings that seniors are more accepting of SST than health providers perceive is valuable for those in clinical practice. I have a few comments on areas of the paper that could use some clarification:

Line 99 Please clarify what you mean by value sets

Lines 143 and 150, What do you mean by “were recruited at arms-length”? Were they recruited via local groups? Was it a convenience sample?

Line 553-554, Suggest adding a few words after “compared with health professional ratings (mean rating 3)” to clarify health professionals are offering ratings of their senior patients. For example, compared with health professional ratings (mean rating 3) of seniors SST usage patterns.

Line 663, Strengths and Limitations. Suggest adding as an additional limitation that the small sample size restricted the ability to breakdown themes by age groups. This prevented analysis of differences in SST usage and acceptance between those in their 70s vs. those in their 80s, for example.

Author Response

(The authors gave the same response as above.)

Reviewer 3 Report

The authors studied the use of health technologies in self-care, considering the perspective of the elderly people and healthcare professionals.

General comments and suggestions for the authors

I suggest considering using “elderly people” rather than “older people”.

The research questions included elderly people with different chronic health condition using a wide range of self-service health technologies, and different healthcare professionals. So, it is difficult to compare the results, especially using a small sample size. This could be included in "Limitations" Section.

Materials and Methods:

2.1. Design: The study was carried out from _____? (month?) to ____?  (month), ____ year?

The participant number could be included in “Results” Section

2.4. Participant recruitment: 

A purposed sampling approach was utilized to recruit participants. Where? Only Sidney?

How long did the authors store the electronic PISCF, and survey and interview data?

The authors could publish a blank questionnaire with the questions and answer options as additional material. Are all the questions mandatory?

2.6 Measurement:

“Data saturation for all questions was achieved by the 7th interview”. This information could be included in “Results” Section. In fact, in “Methods” Section, the authors could write: “The participant number was defined after saturation of responses”.

Results:

3.1. Participant Characteristics

Twenty-two participants answered the online survey, and seven participants agreed to contribute to the interview. However, what was the total number of people who were invited to participate in the survey? The authors could include an estimated number.

The authors described the sociodemographic characteristics of the survey participants. What were the sociodemographic characteristics of  the seven participants who also agreed to be interviewed? The authors could summarize this data in a table.

3.2. Research questions

I suggest to rewrite this section, including tables.

4. Discussion

I suggest including the first paragraph of “Strengths and Limitations” as the first paragraph of the “Discussion” Section.

5- Implications and Conclusions

I suggest to include the implication comments in “Discussion” Section.

Author Response

The rejoinder responds to all reviewer requests, except for the request to use the term 'elderly'. the reason we have not used this term is that it is a labelling term, which older people themselves consider to be derogatory. Modern and preferred parlance in regard to older populations is to denote them as being 'older' rather than being 'elderly'. We have indicated that the study was conducted with Australians. Sydney is the capital city of NSW, which is an Australian state. I think most people in the world would know where Sydney is located, since it is an international city. 

We have responded to all reviewer requests as indicated in the submitted rejoinder

Reviewer 4 Report

This work presents surveys and interviews with people aged 70 and over, and with healthcare professionals, on self-service health technology use in older people using Consumer Dominant Logic. The presented work is well proposed and described properly. This work explores the use of self-service health technology in elderly people and tries to find out the limitations. Outcomes from this study can be helpful for health professionals to train the elderly population in the better use of SST devices. The proposed study can be considered for publication in this journal. However, some suggestions can be incorporated before the publication.

  1. The sample size of the survey may be increased for better and more reliable outcomes.
  2. The survey may include people aged over 60 years.
  3. There are limitations to the SST devices that can be portable. Authors should highlight which type of SST device they have considered while conducting this survey. 

Author Response

We have responded to all reviewer requests as indicated in the submitted rejoinder 

Round 2

Reviewer 1 Report

Abstract.

line59: please move "(n=7)" behind "survey respondents".

Background.

line144: please check references - is ref. 20 be named here by mistake? 

Materials and Methods.

line 256: please correct: "...six month (February 2020 to October 2020)"

line 267: please move "(n=25)" behind "people 70 years and over"

line 268: please move "(n=25)" behind "healthcare professionals"

line 275: There is still a lack of information about the determination of the classification of mild, moderate and severe dementia. As you already pointed out, dementia stages were not tested in advance. Was there at least one corresponding medical diagnosis that can be given here (ICD-10)? If the degrees of severity of dementia were classified without explicit medical diagnosis or testing, how was the classification implemented? If it is not based on an established classification, this should be characterized as a significant methodological limitation that must be pointed out in more detail in the methodology and in the "strengths and limitations"-section than has already been done. If there was an explicit ICD-10 given in advance, please add this information.

Tables 1+2: The tables have been implemented and make it now much easier to capture the topic content. Thank you.

Discussion.

line 1065: please change the order of the references 17 and 4.

Author Response

The rejoinder uploaded addresses all requested amendments.

Reviewer 3 Report

The study was rewritten. However, I suggest some changes to improve it.

The qualitative approach of this mixed study should be well highlighted and described, reinforcing the strength of the study.

Attention to the term “older people” when the inclusion of people aged 70 and over is considered in this work as the starting point of major health issues for most Australians (Lines 186-192)

Abstract:

Line 67: “Since older people are generally willing to use SST’s…”   Are elderly people GENERALLY willing to use technologies? Or the participants of the study?

  1. Materials and Methods

2.1. Design 

Line 255- 256: “This prospective mixed-method study was undertaken over six months (from February 2020 to October 2022)” ….   Please, rewrite the study period (months and years).

 2.3. Participants

Lines 263-265 The number of participants should be described in the results (n=12…. n=10) as already suggested.

2.4. Participant recruitment and consent

The authors answered my questions but did not include them in the article (See below). My doubts could be those of another reader.

Questions: A purposed sampling approach was utilized to recruit participants. Where? Only Sidney? How long did the authors store the electronic PISCF, and survey and interview data? The authors could publish a blank questionnaire with the questions and answer options as additional material. Are all the questions mandatory?

Answers: “All participants were from Sydney, Australia. All data are stored for 5 years after publication of the research. In the survey, all questions were mandatory. All questions will be made available on request to the authors.”

Line 267-268:   “The target population included (n=25) people 70 years and over and (n=25) healthcare professionals”.  …….I suggest writing: “The target population of the quantitative study included (n=25) people 70 years and over and healthcare professionals (n=25)For the qualitative study, the participant number was defined after saturation of responses” (I suggest to move the sentence “the participant number was defined after saturation of responses” from 2.4. to 2.3).

Results:

3.1. Participant characteristics

The authors described the participants of the quantitative study. I reinforce the need to describe those who also contributed to the interviews. Of the 22 participants (quantitative study), who were interviewed? Only on lines 669-671, I could understand that the participants of the qualitative study were 5 elderly people and 2 health professionals, but I don't know if they were doctors, nurses, etc... I suggest to move this information (lines 669-671) to “3.1. Participant characteristics”

We have to remember that the qualitative study explored interesting data.

3.2 Research questions

The tables provided a better organization of the results.

Table 2  

  1. “Personal values influence perceived benefits of health-related SST”. I suggest: Perceived benefits of health-related SST
  2. Sophisticated technological applications inhibit SST adoption. I suggest: Constraints to SST adoption
  3. “Technological challenges need to be addressed”. I suggest: Strategies for better SST adoption and use

  1. Discussion

I reinforce the need to start the conclusion with a summary of the study, considering the strength (mixed methods approach). The results should be summarized (For example: the elderly people of the study showed a willingness to use STTs, but there is still a need for investment in user-friendly technology and engagement of healthcare professionals). 

Author Response

(The authors gave the same response as above.)
